# Hold My Hand: Development of a Force Controller and System Architecture for Joint Walking with a Companion Robot

**DOI:** 10.3390/s23125692

**Published:** 2023-06-18

**Authors:** Enrique Coronado, Toshifumi Shinya, Gentiane Venture

**Affiliations:** 1National Institute of Advanced Industrial Science and Technology (AIST), Tokyo 135-0064, Japan; 2Department of Mechanical Systems Engineering, Faculty of Engineering, Koganei Campus, Tokyo University of Agriculture and Technology (TUAT), Tokyo 184-8588, Japan

**Keywords:** human-robot interaction, force control, robot companion, assistive robotics

## Abstract

In recent years, there has been a growing interest in the development of robotic systems for improving the quality of life of individuals of all ages. Specifically, humanoid robots offer advantages in terms of friendliness and ease of use in such applications. This article proposes a novel system architecture that enables a commercial humanoid robot, specifically the Pepper robot, to walk side-by-side while holding hands, and communicating by responding to the surrounding environment. To achieve this control, an observer is required to estimate the force applied to the robot. This was accomplished by comparing joint torques calculated from the dynamics model to actual current measurements. Additionally, object recognition was performed using Pepper’s camera to facilitate communication in response to surrounding objects. By integrating these components, the system has demonstrated its capability to achieve its intended purpose.

## 1. Introduction

The robotics discipline has rapidly expanded in recent years, not only introducing industrial and collaborative robots (cobots) in factories but also social and service robots to everyday-life scenarios such as restaurants [1], schools [2], shops [3], homes [4,5] and other public spaces [4,6]. Robotic systems in these scenarios are designed to achieve different types of applications, such as care [7], entertainment [8], and education [9]. In many of these applications, researchers explore alternative approaches that propose the use of robots for improving people’s quality of life. A very recent example has been described in [10,11], where a novel type of robot, denoted as “robject”, is designed to strengthen the bond between people and in this way to improve the mental health of potential users. Mental Health is a state of overall psychological well-being [12]. Mental health encompasses all aspects of human cognitive functioning, including thoughts, ideas, motivations, and directions that originate from the mind. These elements have a profound impact on how individuals communicate, behave, and function in both personal and social contexts. By prioritizing mental health, individuals can optimize their well-being and lead fulfilling lives [13]. In fact, ensuring suitable mental health levels is an essential component for maintaining a high quality of life.

Robots have played a crucial role in assisting individuals with autism, dementia, and other cognitive challenges, contributing to the improvement of their treatment. Moreover, these robots have provided valuable companionship to those facing feelings of isolation and loneliness. Additionally, they have been instrumental in enhancing the quality of care and support received by individuals with disabilities from healthcare professionals. Therefore, in recent years, the use of robots to address mental health issues has gained significant attention, particularly to provide solutions that help to mitigate the increase of mental health issues, such as autism [14], dementia [15], and social anxiety [16]. Relevant and recent literature reviews in this area has been presented in [17,18,19]. In this context, one of the primary drivers behind this trend is the growing sense of social isolation and loneliness that many individuals are experiencing [17,20,21]. Moreover, these issues have been recently boosted due to the numerous changes the COVID-19 pandemic has brought to our daily lives. In the case of older adults, who are already at higher risk of social isolation, the pandemic has only intensified feelings of loneliness [22]. This loneliness can exacerbate self-destruction, especially by losing previously active roles and assuming more passive positions [23]. This can lead to a loss of motivation to accomplish activities of daily living (ADL), which can lead to unsafe circumstances and reduced quality of life [24]. However, loneliness is not limited to older adults. Younger generations have also felt the impact of social isolation and disrupted routines. This issue is boosted by the rise in single-person households in countries such as Korea and Japan [25]. As the loneliness pandemic continues to affect all-ages individuals worldwide, the search for technological solutions that can help alleviate its negative mental and physical effects will become increasingly relevant. One possible solution is the use of regular physical activity or exercise to mitigate these negative effects. In particular, moderate exercise plays a significant role in maintaining a healthy lifestyle by offering a range of physical and mental health benefits by releasing endorphins, elevating mood and improving mental well-being [26,27]. Therefore, the development of technological solutions helping to establish a routine and sense of purpose through exercise can potentially combat feelings of isolation, resulting in a safer and more fulfilling life.

Robotics is an emerging technology that has the potential to assist and motivate individuals in their hobbies and entertainment activities [28,29]. Therefore, well-designed robotics applications can represent a suitable solution for enabling individuals to enjoy a better quality of life. However, usability challenges associated with robotics technology can hinder its adoption [30]. These factors or challenges, such as acceptance, attitudes, user-friendliness, trust, utility, safety, anxiety, perceived intelligence, social presence, human-likeness, aesthetics, has been described and reviewed in [30,31,32,33]. In this context, social robots are one of the most promising technologies for assisting people in their daily lives. Compared to other types of robots, social robots are designed to be more approachable, and user-friendly [34]. This facilitates the development of suitable and intuitive applications for the general public [35].

This study aims to develop a system that helps humans to take walks comfortably with a commercial humanoid robot, specifically Pepper robot [36]. To leverage the advantage of the humanoid robot’s familiarity, we aim to implement a Huma-Robot Interaction (HRI) system that behaves similarly to humans when walking hand-in-hand. When two humans walk while holding hands, they need to adjust their direction and velocity depending on the force applied by their partner. To emulate this behavior, we propose a method to estimate the force applied to the robot and a technique to control the robot’s movement based on the estimated force. Moreover, the proposed system can recognize objects and use this information to adapt the robot’s behaviors (speech and gestures).

This paper is organized as follows. Section 2 discusses related work and clarifies the contributions of this article. Section 3 introduces the proposed system architecture enabling a social robot to walk hand-in-hand with a human. Section 4 presents the experimental settings and discusses the obtained results. Conclusion follows.

## 2. Related Work and Contributions

Different robotics devices and solutions have been proposed to assist humans in the last few decades. Examples include meal-assistance robots [37,38,39], brain-computer interfaces controlling wearable robots [40,41], and robot-assisted therapy [42,43]. In this context, some researchers have proposed smart robotics walkers to sustain the mobility of elderly people, or with some motion disabilities [44,45,46]. Smart robotics walkers “intend to assist the mobility function of disabled people that present reduced lower motor function and low balance, by improving their autonomy” [47] by facilitating a stable gait and easy maneuverability. However, the focus of this article is not to support mobility. Instead, we propose a system architecture enabling an industrial-produced social robotic platform to walk hand-in-hand with humans. The persuasive and motivational capabilities of robots are particularly relevant in various cases, especially those involving people’s health recovery and maintenance [48]. Numerous studies have demonstrated the effectiveness and practicality of robotics systems in motivating people during rehabilitation [49,50] as well as in promoting healthy activities like exercise [51,52] across different age groups. The application discussed in this work shares similar objectives by developing a system architecture that can potentially be used to promote a healthy lifestyle through walking. Therefore, the development of a smart robotics walker device for individuals with motion disabilities falls outside the scope of this work.

Despite the increasing popularity of social and humanoid robots, few researchers have proposed solutions for comfortable physical human-robot interaction (pHRI) during walking. In this context, Granados et al. [53] proposed a mobile humanoid robot that uses pHRI to lead humans during dance training. These dances have various motion directions, including forward and backward walking. The authors of [54] recently presented the Ibuki actuated child-like android and an application that enables walking hand-in-hand with this robot. This application allows the user to pull on Ibuki, causing it to change direction and follow the user closely. They also developed a behavior where Ibuki can lead a person by holding their hands. For this, the Ibuki robot is equipped with an internal sensor, specifically a potentiometer embedded at the shoulder. This sensor is used to detect changes in the shoulder roll joint angle. In this way, the robot can detect when the user pulls on the robot’s arm. Additionally, a Laser Range Finder (LRF) was implemented in Ibuki to avoid obstacles during movement. In fact, walking hand-in-hand with a robot requires highly sophisticated and advanced interaction skills. In this scenario, a crucial skill that a robot needs to possess is the ability to respond to the partner’s movements or applied force and adapt its movements accordingly. Moreover, the robot needs to have a suitable situational awareness level to help humans avoid obstacles or create more engaging interactions. However, few commercially available or industrial-produced social robots are sophisticated enough to meet these requirements.

Pepper [36], developed by the French-Japanese company Aldebaran/Softbank Robotics, is a sophisticated social robot popular and accessible in some countries, such as Japan. Pepper has been designed to interact with humans in various contexts, such children robot interaction [8], rehabilitation care [55] and public spaces [56], making it a suitable candidate for enabling humans to walk hand-in-hand through pHRI. In [57], a solution enabling Pepper to act as a walking trainer for elderly people and rehabilitation patients is proposed. Their authors use a wheel compliance approach to match the motion intention of patients with the robot’s pace. For this, they utilize a method that does not require sensors to detect soft pressure on the robot’s surface. This enables them to evaluate the difference between the external and required displacements necessary to keep the patients in a stable position. Then, the robot’s wheels receive the direction and intensity commands that enable the robot to match the user’s speed. However, patients must lean on the robot, which can cause overheating of Pepper’s knee joint due to the substantial amount of weight that Pepper must support. As noted in [57], this issue can sometimes result in the system needing to be restarted. To the best of the authors’ knowledge, only a limited number of research works have explored Pepper’s capabilities for creating a hand-in-hand walker companion. Notable studies, such as [58,59], are among the few that have investigated this area. However, the main target of both studies was primarily Human-Child Interaction. In both studies, children can pull the robot’s hand in the desired direction of movement. The robot uses a direction recognition algorithm that compares the current position of the end effector (left hand) and a default pose. For example, when a child pulls the robot forward, it moves forward in response. Similarly, when the child pulls the robot backward, it moves backward. If the child pulls the robot’s hand to the right, the robot rotates clockwise, while pulling the hand to the left causes the robot to rotate counterclockwise. Moreover, the velocity of the movements varies according to the height of the left arm. Our approach differs significantly from the studies by Kochigami and colleagues in [58,59], as we use the force, rather than hand position, applied to the Pepper robot to control the Pepper’s direction and velocity of movement.

We summarize the contributions of this article as follows:Development of a system architecture for pHRI that allows humans to walk while holding hands with Pepper robotIntegration of a vision-based interactive system that recognizes objects in the environment and adapts the robot’s behaviors accordingly.

## 3. System Architecture

This article presents a system architecture for pHRI enabling a Pepper robot to walk with humans while holding hands. Pepper has physical dimensions comparable to those of a primary school child, with a height of 1210 mm, a width of 480 mm, a depth of 425 mm, and a weight of 29 kg. Pepper consists of a head, torso, two arms, and one leg. Moreover, Pepper is standing on a holonomic mobile base that is equipped with three omnidirectional wheels [60]. These wheels allow Pepper to move forward, backward, left, and right and rotate left and right. Pepper has 20 degrees of freedom throughout its body, allowing it to replicate human-like movements. When operating normally, Pepper has a function called Autonomous Life. This operational mode was designed to create a greater familiarity with people by replicating movements similar to human breathing and enabling the robot to track people’s gaze, among other things. In addition, Pepper is equipped with a feature that stops the movement of its arms and body when objects are detected in its vicinity using infrared sensors, bumpers, and cameras. For our research, we disabled the Autonomous Life mode and the stop feature since they could affect force estimation and navigation while standing close to the robot.

To enable the Pepper robot to walk while holding hands with a person, the robot’s movements must be synchronized with the person’s movements. In this study, the person’s relative position is estimated by analyzing the force exerted by the person’s hand on the robot’s hand. Based on this estimation, the robot’s speed and direction are adjusted to ensure smooth and coordinated walking. Figure 1 shows the proposed system architecture for achieving this goal. It consists of four main sub-systems: (i) a data acquisition sub-system used to get information from sensors of Pepper, (ii) an object recognition interface used to provide basic situation awareness, (iii) a Python 3 script implementing the Robotic Toolbox [61] used for estimating the exerted forces on the robot, (iv) and a cognition and control module enabling Pepper to adapt its behaviors. More details about the functionalities of these sub-systems are explained below. In this study, we established connectivity between Python 2 and Python 3 modules using the NEP libraries [62], designed by our research team members. Our goal in selecting this approach was to create a system architecture that is accessible to different types of users. For this, NEP libraries are designed to be cross-platform, ensuring that other researchers can easily replicate and utilize our system, regardless of their preferred platform.

The general control algorithm, depicted in Figure 2, enables the robot to walk hand-in-hand with a human while interacting with the environment. In this particular scenario, the force estimation is performed only when the user is holding the robot’s left hand, allowing the human to guide the robot’s movements. Concurrently, the object recognition system facilitates simple interaction with the environment. It is important to note that object recognition is not utilized for obstacle avoidance purposes. If a new object is detected after a predefined time interval, the robot stops and suggests an action to the human. Otherwise, it continues to follow the human’s guidance.

### 3.1. Data Acquisition

Pepper is equipped with different sensors that enable it to respond to its surroundings. In this study, we primarily used the joint angle and current sensors implemented in Pepper to monitor the status of the motors in each joint. This information is crucial for the control strategy proposed in this article. We also employed contact sensors in the foot to initiate and reset the robot application. To acquire data from these sensors, we utilized a Python 2 script that runs on the Pepper Software Development Kit (SDK) (http://doc.aldebaran.com/2-5/dev/python/index.html, accessed on 20 May 2023). Additionally, we used images from the Pepper camera to recognize objects in the environment. For this purpose, we developed an interface (shown in Figure 3) to capture images from the Pepper robot. The interface runs on a Node.js (https://nodejs.org/en/, accessed on 20 May 2023) application that executes Python scripts with the necessary arguments (e.g., Figure 3 demonstrates how to set up the Pepper camera to obtain images at a resolution of 320 × 240). The executed Python 2 script by this interface uses the OpenCV (https://opencv.org/, accessed on 20 May 2023) and Pepper SDK frameworks to get and stream images from the Pepper robot. These images are published to the network and read by the Object Recognition module.

### 3.2. Object Recognition

We developed an additional interface on top of Node.js to set up and launch a Python 3 script running the YOLO (You Only Look Once) algorithm. This user interface, shown in Figure 4, allows the selection of the trained model to use the topic where the robot images are published, and the image size resolution for processing using YOLO (e.g., by selecting yolo-320 a resolution of 320 × 320 is set). Using smaller image sizes in YOLO models can speed up object recognition processes but may result in lower accuracy. The Object Recognition modules output a list of recognized objects, their confidence scores, and their bounding box positions (origin, height, width, and center). With this interface, the proposed system can detect objects of interest and provides valuable data to adapt the robot’s behaviors. A pre-trained deep learning model using YOLO is composed of three main files: a .cfg file that defines the deep neural network architecture and parameters, a list .names file with the list of the possible classes to recognize, and a .weight file with the values of weights of the neural network. These files must be saved in a folder with a name that is easy to recognize (in the figure defined as yolo4) on a specific path of the computer where several pre-trained models can be located. The location of this specific and default path depends on the operating system in which the interface is executed. In the case of Windows, the path is “C:/nep/deep_models/objects”. This interface is able to locate the models saved on the default path and set the arguments that enable the Python 3 script running the YOLO to load the required files.

### 3.3. Force Estimation Using Python Robotic Toolbox

Pepper does not have force nor torque sensors. Therefore, we estimate the force exerted by the human partner on Pepper’s arm using a dynamic model. We used the Robotics Toolbox Python library to calculate the forces applied by humans on Pepper when holding hands. This library, developed by Peter Corke et al. [61], specializes in robot-related calculations, such as kinematics and dynamics, and facilitates the creation of robot models through the specification of their Denavit-Hartenberg (DH) parameters. Additionally, it can be used to create 3D representations of robot models, allowing visualization of their movements in simulations. In this work, we used the Robotics Toolbox to create Pepper’s dynamic model and calculate the torques required for posture control. Moreover, the Jacobian used for force estimation calculations was also computed using the Robotics Toolbox. For simplicity, Pepper was modeled as a 9-link robot from its foot to left hand in the model, as shown in Figure 5. DH parameters used in the creation of the Pepper models using the Robotics Toolbox Python library are shown in Table 1. The coordinate system of Pepper and force the directions of the applied force on Pepper’s left hand are shown in Figure 6.

To estimate the force exerted by the human partner on Pepper’s arm, we use the dynamic model expressed by the following equation:(1)$$\tau =M\left(\theta \right)\ddot{\theta }+C\left(\theta ,\dot{\theta }\right)+G\left(\theta \right)+h\left(\dot{\theta }\right)$$

Equation (Equation 1) describes the relationship between the driving torques (τ) and the joint angles (θ). In this equation, M represents the inertia matrix, C is the vector of Coriolis and centrifugal torques, G is the gravity torque vector, and h is the vector of friction torques. When walking while holding hands, the robot arm’s movement is mainly at a constant speed, with few sudden changes in posture. Thus, it is assumed that the robot arm’s movement during walking is nearly static, and the joint velocities ($\dot{\theta }$) and joint accelerations ($\ddot{\theta }$) are close to zero. Therefore the influence of the term $M\left(\theta \right)\ddot{\theta }$, $h(\dot{\theta })$, and $C(\theta ,\dot{\theta })$ is considered small, and ignored in the calculation. We estimate the actual driving torque from the readings of the current sensor and the motor constants. By applying the values obtained from the joint angle values of motors to Equation (Equation 1), the torque required for posture control can be derived from the dynamic model. When an external force is exerted on the robot arm, the difference between the torque calculated by the dynamic model and the actual torque generated by the external force is referred to as $\tau_{d}$. The torque applied to Pepper’s hand by the human is calculated using the estimated torque values obtained from Equation (Equation 1). According to the principle of virtual work, the following equation is derived:(2)$$f_{d}={\left(J^{T}\right)}^{-1}\tau_{d}$$
where $f_{d}$ represents the external force and $\tau_{d}=\hat{\tau }-\tau$ is the difference between the estimated driving torque $\hat{\tau }$ and the actual torque τ. The J represents the Jacobian matrix, which relates the change in end effector position Δx to small changes in joint angles Δθ as follows:(3)Δx=J(θ)Δθ

The actual operating torque is calculated by obtaining the electric current value from the current sensor installed on each of Pepper’s motors and multiplying it by the motor constant. The motor constants used for calculating the actual torque in Pepper are shown in Table 2. However, the current sensor installed in Pepper can only obtain absolute values and cannot determine the direction of the current. Therefore, the direction of the actual operating torque obtained from this current was assumed to be the same as that of the estimated drive torque.

The Jacobian matrix can be calculated from the link parameters and joint angles. Since the created model of Pepper has 9 degrees of freedom, the Jacobian matrix becomes an irregular matrix of size (9 × 3). As a result, an inverse matrix does not exist, so the pseudo-inverse matrix $J^{†}$ of the Jacobian matrix is used for the actual calculations instead.
(4)$$f_{d}={\left(J{\left(\theta \right)}^{T}\right)}^{†}\tau_{d}\;$$

For building the dynamic model of robot, we used the values of mass, center of mass (CoM), and inertia matrix $I_{0}$ from the official webpage of the Pepper robot (http://doc.aldebaran.com/2-0/family/juliette_technical/masses_juliette.html, accessed on 30 May 2023).

The Robotics Toolbox is only available to be executed in Python 3, and the Pepper SDK can only be executed in Python 2. Therefore, we use the Python libraries of the NEP framework, which support both Python 2 and Python 3, to send the outputs of the force estimation module to the robot cognition and control module executing the Pepper SDK.

### 3.4. Robot Cognition and Control

This module is in charge of sending the control commands that control the direction and speed of the robot platform based on the force detected. Moreover, this module receives the values of the object recognition module to adapt the robot’s behaviors according to the status of the environment. The robot behaviors adapted by this module are speech and gesture. To execute these behaviors, we used the functionalities provided by the Pepper SDK. An example of implemented behaviors is enabling Pepper to draw the user’s attention to objects of potential danger in their surroundings or of potential interest. To accomplish this, the robot makes arm and head movements in the direction of the detected object while providing verbal communication. In a second example, Pepper can suggest taking a sit when a chair is detected. For this, the robot can obtain the coordinates of the recognized object in the image and make individualized movements accordingly. Figure 7 provides a visual representation of how Pepper points to the position of detected objects. In this application, speech is performed when Pepper stops its movement. Moreover, two types of intervals were set, one for not reacting to the same object for a certain period and the other for not performing a speech act for a certain period. These intervals prevent the robot from reacting too frequently to objects and stopping too often.

The Pepper robot is equipped with tactile sensors located on its palms and fingertips. These tactile allow the robot to sense when someone is holding its hand. Once the sensors detect the pressure, they send signals to the robot’s internal system enabling it to interpret the touch and respond appropriately. In this work, the control algorithms are executed only when the robot’s internal system detects that the Pepper’s hand is touched. Therefore, the estimation of the force is not executed when the user is not holding the robot. As a result, force estimation is not performed when the user is not holding the robot, which prevents the robot from unintentionally navigating in an unexpected direction.

## 4. Experimental Validation

This section presents the experimental results from an initial test of our proposed system architecture conducted in an open and public coffee space room adjacent to the GVlab at the Tokyo University of Agriculture and Technology. As depicted in Figure 5, this public space is primarily obstacle-free. It is important to note that in this particular application, the human assumes control of the robot, and therefore, no obstacle avoidance features were implemented. A video proving the technical suitability of our proposed system architecture in a 3 min experiment is available online (https://youtu.be/aVKmIPbaSO8, accessed on 30 May 2023). Figure 8 shows some captures from this video. This figure shows how our proposed system can accurately detect objects in the environment using images obtained from Pepper’s camera. Moreover, Figure 9 shows an example of the values obtained from the force estimation module when applying a forward force on the Pepper robot. The plot shown on the left of Figure 9 represents the real-time results from the module using the Robotics Toolbox to calculate the applied force to the robot.

### 4.1. Object Recognition

In this test, object recognition was carried out using images captured by Pepper’s camera to detect different objects, such as “refrigerator”, “chair”, “bottle” and “monitor”. These objects were placed between 2 m and 4 m from Pepper, as shown in Figure 10. Pepper SDK provides the ALVideoDevice to facilitate image acquisition by establishing a Wi-Fi socket connection between Pepper’s internal computer and an external computer (in this case, where object recognition is performed) through Wi-Fi. For using ALVideoDevice features, a suitable resolution and color space must be set to use this module. In this scenario, the framerate of obtained images is highly dependent on the resolution defined in the ALVideoDevice module. For this application, it was essential to select the resolution carefully in order to provide acceptable accuracy levels in the object recognition system. Figure 10 depicts object recognition results using a 640 × 480 pixels resolution. Our findings indicate that the recognition of objects was successful if the object occupied about a quarter of the image. As shown in Figure 10, objects with a size of approximately 80 cm can be detected with high accuracy up to a distance of 4 m. However, the object recognition system also misidentified a training machine (not in the learning dataset) on the right side as a monitor. In Figure 11, we present object detection results at 2 m and 4 m distances using a 320 × 240 pixels resolution. We observed that the accuracy of the object recognition system significantly decreased in this resolution, even at close distances, compared to the high-resolution case. To improve the accuracy of our system, we use the reliability score obtained by the object recognition module. In the case of the training machine, which was mistakenly recognized as a monitor in Figure 10, the reliability score was smaller and unstable compared to the chair recognized at the same time, and it frequently changed between the recognized and unrecognized states. Therefore, to reduce the misrecognition of objects in the walking assistance system, we exclude objects with low average reliability scores or those recognized with low-reliability scores. Furthermore, objects with ambiguous outlines, such as bottles, can be difficult to detect, resulting in low-reliability scores. To address this issue, we established a different threshold for the reliability score for each object. However, it’s important to note that these values may vary depending on different scenarios, camera resolutions, and illumination conditions.

The novelty of this work does not involve the training and analysis of specific object recognition methods or algorithms. Instead, we propose a system architecture that offers a user-friendly interface, allowing the loading of any pre-trained YOLO-v4 model using OpenCV and TensorFlow. Table 3 presents the list of objects available for recognition using the selected pre-trained model. The choice of this specific pre-trained model was based on its inclusion of common objects found in indoor scenarios.

According to the results of object recognition, short utterances were spoken along with gestures. An example of a spoken utterance is shown below. If multiple lines are assigned to a single object, the utterances are spoken sequentially as they are recognized.

“chair”: “There is a chair. Do you need a rest?”“chair”: “There is a chair again. Don’t you still need a rest?”“person”: “Hello. It’s a good day for walking”“laptop”: “There is a laptop. Someone is working hard on the research”“bottle”: “Oh, there is a bottle. We can’t buy them in TUAT”“bottle”: “I found the bottle again. Waste should be thrown away”“bottle”: “Would you like to go get something to drink?”“cell phone”: “I found a cell phone. Did someone leave it?”

### 4.2. Force Estimation

To evaluate the suitability of force estimation, we conducted an experiment where a person applied forces in six directions (forward, backward, left, right, up, and down) with varying magnitudes to the Pepper robot’s hand while fixed in a specific posture. The results of force estimation for the *x*, *y*, and *z*-axes are shown in Figure 12, where the *x*-axis points forward, the *y*-axis points to the left, and the *z*-axis points vertically upward. We observed that the reading values in the direction of the applied force proportionally increased when force was applied to the corresponding direction, indicating the system’s ability to measure both the magnitude and direction of the force. However, we observed errors in some cases, particularly when forces were applied along the *x* or *y*-axis. For instance, as seen in Figure 12, an applied force in the x negative direction is detected when the force is mainly applied in the z positive direction. Moreover, when force was applied in the positive x-direction or the positive and negative y-directions, we observed significant components of force in the z-direction and the x-direction, respectively, even though no force was being applied in those directions. This can be attributed to the fact that the force applied by the human hand is not purely axial. Moreover, we noticed that errors in the readings of angles and torques increased as the joints exceeded their operating range. In this work, we propose a method that uses the force estimation results to regulate Pepper’s walking speed while holding hands with a human. However, only the force estimation results in the x and y directions are employed to adapt Pepper walking. Thus, the effect of the significant errors in the force estimation results along z is considered negligible for this particular application. Also a precise force measurement is not necessary, the direction and rough amplitude are sufficient for this application, therefore the proposed force observer is sufficient.

### 4.3. Limitations

Our tests suggest that the estimated force can be accurately detected, providing a smooth feeling when walking while holding hands with the Pepper robot. However, there were some cases where the force control accuracy could be reduced. We observed that this issue is presented due to two specific reasons. On the one hand, we observed that the results of force estimation were affected by Pepper’s slight posture changes and variations in current values. This can result in frequent changes in speed, leading to unnatural movements. On the other hand, errors in force estimation are also present when the body shakes over small asperity on the ground or when the arm is suddenly pulled strongly. Additionally, Pepper is designed as a robot with no branching from its legs to its left hand in the dynamics model, so errors in force calculation may occur due to changes in the posture of the right hand or head, causing Pepper’s center of gravity to shift. In our system, when significant discrepancies occur in force estimation during force control, parameters of the force control can be readjusted by pushing Pepper’s bumper at the foot.

Since the main objective of this study is to demonstrate the technological suitability of the system, no participants were recruited for experiments. The system was tested by the authors of this article. As a result, this research does not require consent from any individuals, as no human subjects (aside from the authors of this article) were involved in the study. The involvement and recruitment of external participants will be considered for future work, which can potentially involve the analysis of usability and user experience factors. Furthermore, these studies can be expanded to encompass various types of soil surfaces on which Pepper walks or different interaction scenarios.

## 5. Conclusions

This article presented a novel software architecture enabling an industrial-produced humanoid robot to walk hand-in-hand with humans using a pHRI approach. The proposed systems integrate object recognition skills to enable the robot to react to the surrounding environment. Due to the lack of force sensors in Pepper, an estimation of the force applied by the human hand to the robot hand was required. The force estimation was performed using dynamic simulation based on the difference between actual and estimated joint torque. Although we were able to create a suitable system for our purpose, some issues arose due to the assumptions made during modeling in the simulation, which resulted in some problems in certain cases where there were large changes in posture.

The walking speed of humans can vary based on factors like age, fitness level, terrain, and purpose of movement. While the maximum velocity of the Pepper robot may be slower than a healthy human’s typical walking speed, it’s crucial to consider that the robot’s primary objective is not to replicate human locomotion but to engage with humans in social and service contexts. When walking hand in hand with other humans, the average velocity tends to be slower, influenced by individual walking speeds, physical condition, and comfort level. This slower pace offers opportunities for social bonding, conversations, shared experiences, and a sense of togetherness. Individuals naturally adjust their pace to accommodate slower walkers, ensuring the group remains connected. In this context, future research directions could investigate potential discomfort or impatience, as well as the impact on social bonding, by considering different walking velocities and examining how the Pepper robot’s slowdown in walking pace during hand-in-hand interactions affects human partners of various age groups.

Another important aspect to consider in future work is the influence of Pepper’s structure on the walking activity. This includes not only the shoulder angle but also the size and height of individuals. Children, with their shorter height and smaller size, are more prone to potential collisions with the robot’s base during walking. They may encounter challenges in maintaining a safe distance and may not fully comprehend the associated risks. Additionally, children tend to have less awareness of their surroundings and exhibit more unpredictable movements, further increasing the likelihood of accidental contact with the robot’s base. To mitigate these collision risks, the utilization of Pepper’s range sensors becomes crucial. By leveraging the range sensor data, the robot can effectively detect obstacles and potential collision hazards, including objects and body parts that may come into contact with its base. This enables the robot to adapt its walking trajectory, slow down, or stop altogether when necessary to avoid accidents.

Finally, as part of our future work, we aim to enhance Pepper’s capabilities by incorporating additional vision-based skills, such as face and speech recognition. These advanced functionalities would enable Pepper to exhibit more sophisticated and interactive behaviors.

## Figures and Tables

**Figure 1 sensors-23-05692-f001:**
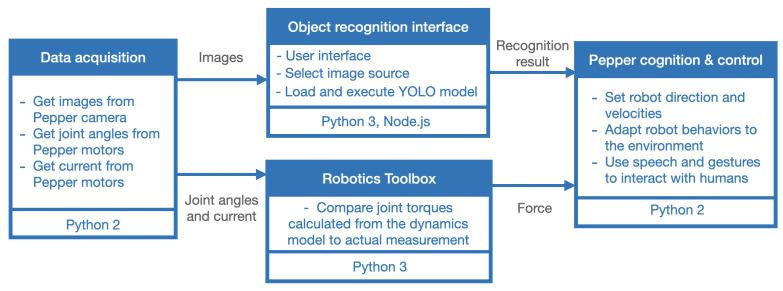
Proposed system architecture.

**Figure 2 sensors-23-05692-f002:**
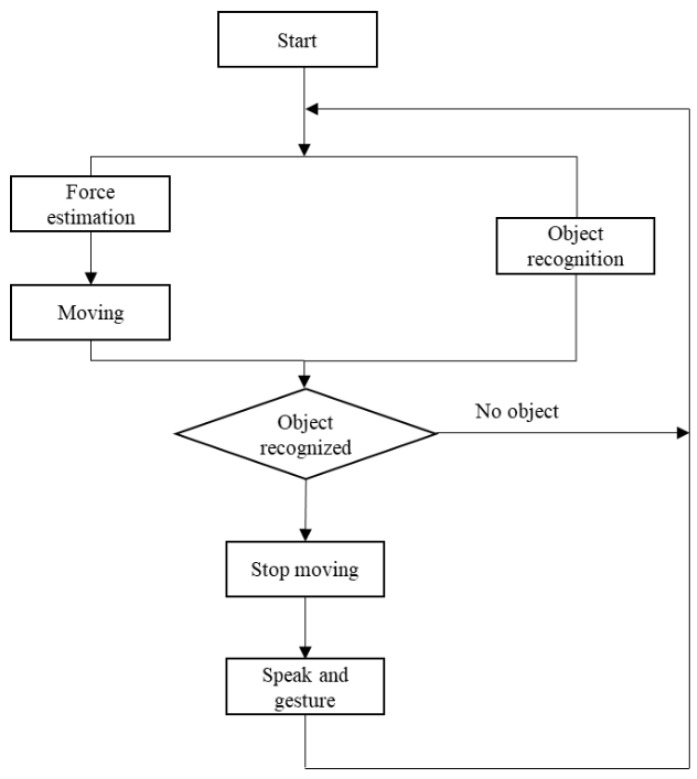
General control algorithm integrated in this work.

**Figure 3 sensors-23-05692-f003:**
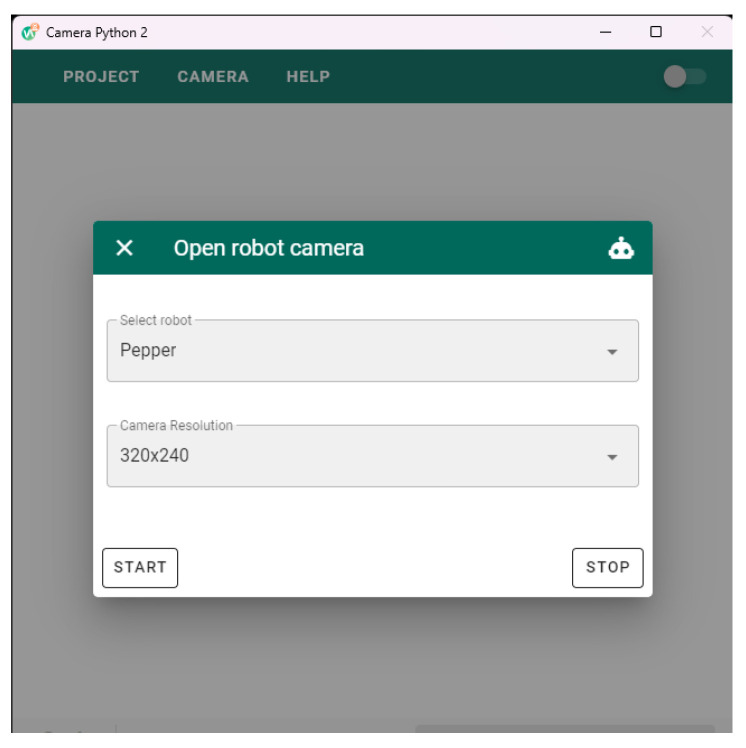
Camera acquisition interface used to collect and publish images from the Pepper camera to the network. The user can also select the resolution of the images obtained. Lowering the resolution of images obtained from Pepper’s camera reduces the latency between the images.

**Figure 4 sensors-23-05692-f004:**
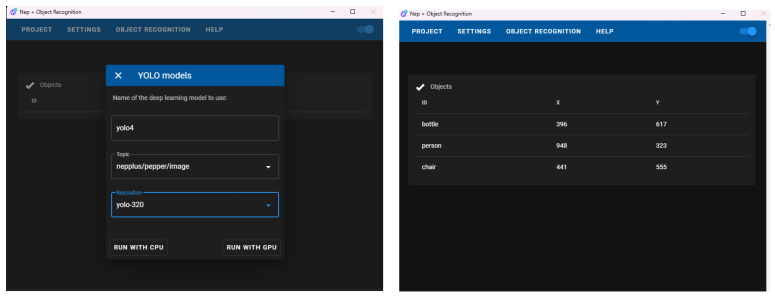
Recognition application developed to enable Pepper to recognize objects in its environment using the YOLO algorithm. On the interface shown on the left side, users can select the parameters and pre-trained deep learning model to use, while on the right side, the user can see which objects are detected from the camera of the Pepper robot.

**Figure 5 sensors-23-05692-f005:**
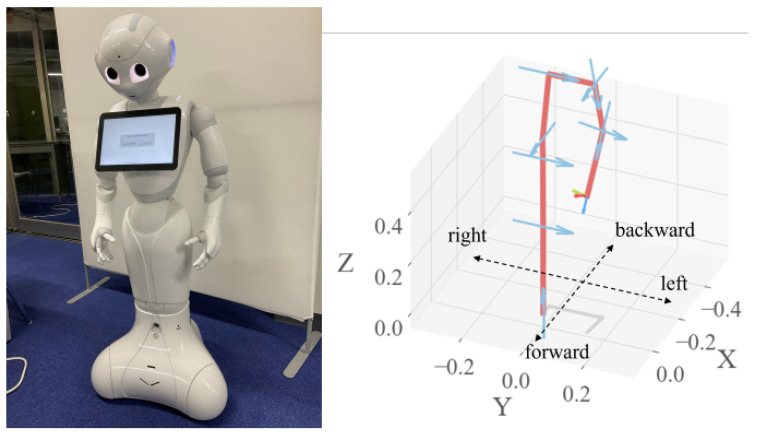
Pepper robot and their geometric model represented as a 9-link mechanism that includes the elements from its foot to left hand.

**Figure 6 sensors-23-05692-f006:**
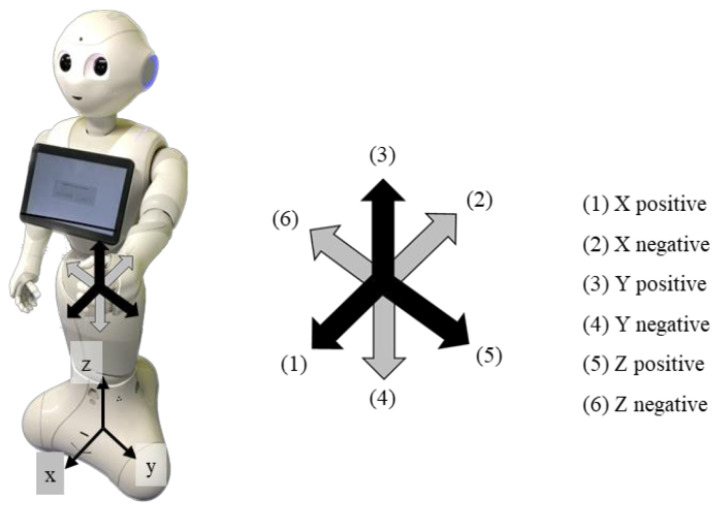
Coordinate system of Pepper. The thin arrows indicate the directions of the applied force on Pepper’s left hand.

**Figure 7 sensors-23-05692-f007:**
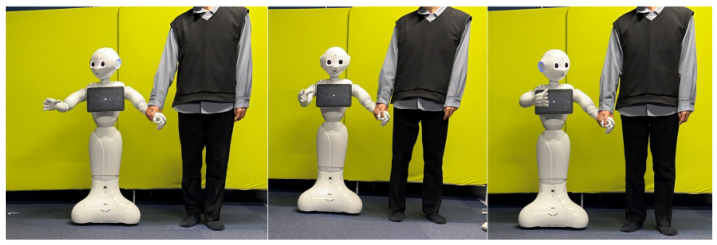
Gesture of looking and pointing at the recognized object.

**Figure 8 sensors-23-05692-f008:**
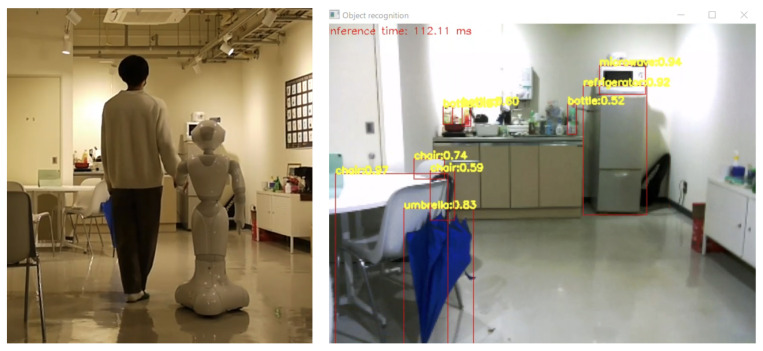
Example of object recognition results when walking with Pepper robot in a public space.

**Figure 9 sensors-23-05692-f009:**
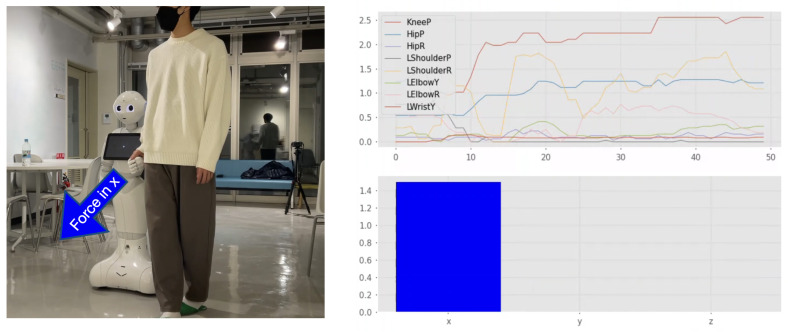
Example of force results when walking with Pepper robot. The module using the Robotics Toolbox is used to plot the current values of each joint and results of the force estimation method.

**Figure 10 sensors-23-05692-f010:**
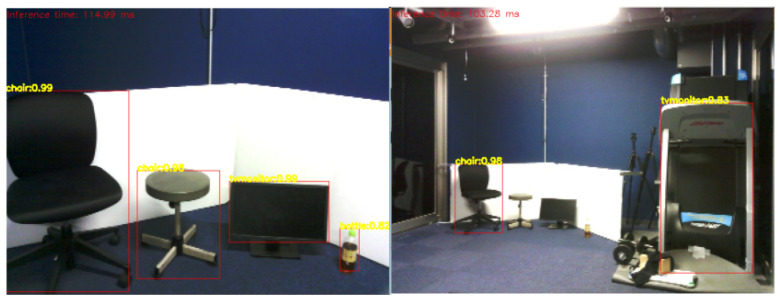
Result of object recognition (left: 2 m, right: 4 m).

**Figure 11 sensors-23-05692-f011:**
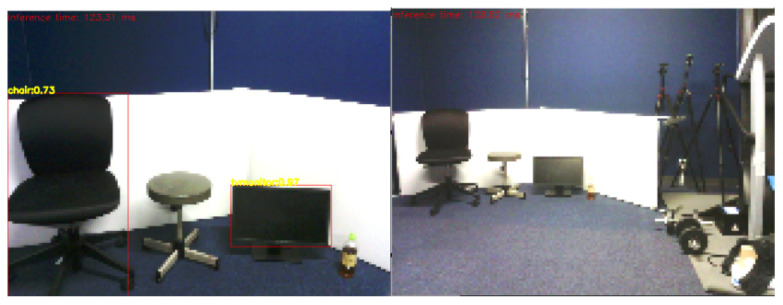
Result of object recognition in low resolution (left: 2 m, right: 4 m).

**Figure 12 sensors-23-05692-f012:**
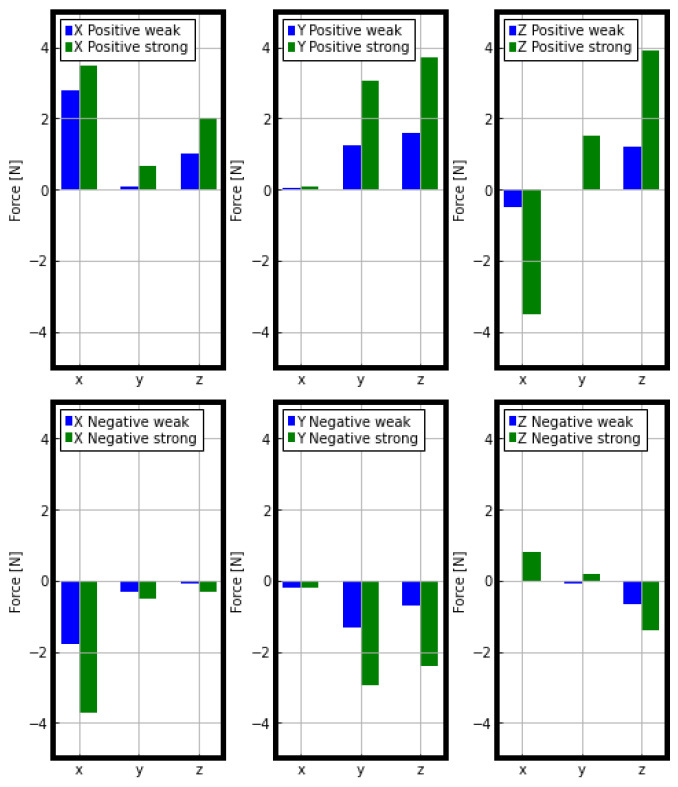
Result of force estimation experiment.

**Table 1 sensors-23-05692-t001:** Denavit-Hartenberg parameters of Pepper robot.

Joint	$a_{i}$	$\alpha_{i}$	$d_{i}$	$\theta_{i}$
Base	0	$-\frac{\pi }{2}$	339	0
Knee Pitch	268	0	0	$-\frac{\pi }{2}$
Hip Pitch	79	$-\frac{\pi }{2}$	0	−57
Hip Roll	$-\frac{\pi }{2}$	0	181	0
Head Yaw	0	$-\frac{\pi }{2}$	0	150
Head Pitch	0	0	1	150
Right Shoulder Pitch	0	$-\frac{\pi }{2}$	0	181
Right Shoulder Roll	226	$\frac{\pi }{2}$	1	0
Right Elbow Roll	0	$-\frac{\pi }{2}$	150	0
Right Elbow Yaw	226	$\frac{\pi }{2}$	0	$-\frac{\pi }{2}$
Right Wrist Yaw	0	0	0	0
Left Shoulder Pitch	0	$-\frac{\pi }{2}$	181	0
Left Shoulder Roll	226	$\frac{\pi }{2}$	0	$\frac{\pi }{2}$
Left Elbow Yaw	1	$\frac{\pi }{2}$	150	0
Left Elbow Roll	226	$-\frac{\pi }{2}$	0	$-\frac{\pi }{2}$
Left Wrist Yaw	0	0	1	0

**Table 2 sensors-23-05692-t002:** Motor torque constants of Pepper.

Joint	Motor Torque Constant mN · m/A
Knee Pitch	36.9
Hip Pitch	36.9
Hip Roll	47.5
Shoulder Pitch	27.5
Shoulder Roll	19.2
Elbow Yaw	27.5
Elbow Roll	19.2
Wrist Yaw	20.1

**Table 3 sensors-23-05692-t003:** Detectable objects.

Index	Object	Index	Object	Index	Object
0	person	17	horse	34	baseball bat
1	bicycle	18	sheep	35	baseball glove
2	car	19	cow	36	skateboard
3	motorbike	20	elephant	37	surfboard
4	airplane	21	bear	38	tennis racket
5	bus	22	zebra	39	bottle
6	train	23	giraffe	43	wine glass
7	truck	24	backpack	44	cup
8	boat	25	umbrella	45	fork
9	traffic light	26	handbag	46	knife
10	fire hydrant	27	tie	47	spoon
11	stop sign	28	suitcase	48	bowl
12	parking meter	29	frisbee	49	banana
13	bench	30	skis	50	apple
14	bird	31	snowboard	51	sandwich
15	cat	32	sports ball	52	orange
16	dog	33	kite	53	broccoli

## Data Availability

Not applicable.

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
