# Peer review of "Hold My Hand: Development of a Force Controller and System Architecture for Joint Walking with a Companion Robot"

_sensors, 2023, doi:10.3390/s23125692_

Round 1

Reviewer 1 Report

This article seems interesting, but needs to improve in some aspects:

It seems a report describing the capabilities of the commercial robot.

That is why the next recommendation is to improve it. (Please consider).

1. Lack of mathematical information. (need the dynamic model terms of the human-robot)

2. Equation (2) is not well defined.

3. There is no force control algorithm. (Please include an algorithm)

Author Response

General comment:

This article seems interesting, but needs to improve in some aspects:

It seems a report describing the capabilities of the commercial robot.

That is why the next recommendation is to improve it. (Please consider).

Response: Thank you for the feedback. To tackle this issue we added more implementation details.

Comment 1:

  1. Lack of mathematical information. (need the dynamic model terms of the human-robot)

Response: 

  • We added Table 1 , which give information about the Denavit-Hartenberg parameters of Pepper robot used in the Python robotics toolbox
  • We added Table 2 , which give information about the motor torque constant of Pepper. 
  • We added link to the relevant dynamic model terms of Pepper robot, such as mass, center of mass, and inertial matrix (Line 274 and 275)

Comment 2:

  1. Equation (2) is not well defined.

Response: We rewrite sentences explaining equation 2 (lines 256 to 262)

Comment 3:

  1. There is no force control algorithm. (Please include an algorithm)

Response: 

We added the lines 184 to 191 and the figure 1 to explain the general control algorithm implemented in this work.

The general control algorithm, depicted in Figure 1, enables the robot to walk hand-in-hand with a human while interacting with the environment. In this particular scenario, the force estimation is performed only when the user is holding the robot's left hand, allowing the human to guide the robot's movements. Concurrently, the object recognition system facilitates simple interaction with the environment. It is important to note that object recognition is not utilized for obstacle avoidance purposes. If a new object is detected after a predefined time interval, the robot stops and suggests an action to the human. Otherwise, it continues to follow the human's guidance

Reviewer 2 Report

The authors propose a novel system architecture that enables a commercial humanoid robot, specifically the Pepper robot, to walk side-by-side while holding hands, and communicating by responding to the surrounding environment. This is done by comparing joint torques calculated from the dynamics model used and to actual current measurements using the Python Robotic Toolbox. The idea is very interesting and the paper is well writen with a few minor errors detected. The next comments are intended to improve even more paper quality and readers' understanding.

In order to make the work more reproducible, please provide more detail about how the estimations were implemented using the Python Robotic Toolbox. Thank you for the link to the video including the demonstration of the robot in operation, it was very important and helped to clarify how the proposed solution works.

How do you detect that the user is holding the robot's hand/arm? Do you assume they are always in contact? Would it be possible for the calculation to estimate force even when the user is not holding the robot, causing the robot to wrongly navigate to a certain direction?

The authors claim that the work is not intended to people with motion disabilities nor is it intended to support mobility. Therefore, I may assume that the solution is destined to healthy users. Since there was no limitation on the age of the possible users, walking speed may vary from a slower pace to a faster one. Did you perform any analysis on the maximum speed of the robot so that it would be possible to perceive if the robot is "slowing down" the human walking pace? It seems that the proposed solution would be more adequate to users with higher age, as the walking pace diminishes as the person gets older.

Another important point that may be already included in the authors plans for future work (usability analysis) is the influence of Pepper's structure in the walking activity. Due to the format of its base, the user's feet/legs may touch the robot's base causing some accident. One possible solution is to increase the shoulder angle to make the robot a little bit more distant to the user. Do you believe is there any impact in the estimation of the forces if the user is holding the robot arm in a distance higher than the one showed on the demo video?

More general comments and minor errors are listed as follows.

"an Huma-Robot" -> "a Human-Robot"

"use, the topic" -> "use the topic"

"Moreover, 7" -> "Moreover, Figure 7"

" figure 8 objects" -> " figure 8, objects"

Author Response

General comment:

The authors propose a novel system architecture that enables a commercial humanoid robot, specifically the Pepper robot, to walk side-by-side while holding hands, and communicating by responding to the surrounding environment. This is done by comparing joint torques calculated from the dynamics model used and to actual current measurements using the Python Robotic Toolbox. The idea is very interesting and the paper is well writen with a few minor errors detected. The next comments are intended to improve even more paper quality and readers' understanding.

Response: Thank you for the positive feedback. We have taken the comments of the reviewer into account to improve the quality of our paper and enhance readers' understanding.

Comment 1: 

In order to make the work more reproducible, please provide more detail about how the estimations were implemented using the Python Robotic Toolbox. Thank you for the link to the video including the demonstration of the robot in operation, it was very important and helped to clarify how the proposed solution works.

Response:

We added some more details about how calculations were done using the Python Robotic Toolbox in a Python script. The Robotics Toolbox facilitates the construction of robot models using Denavit–Hartenberg (DH) parameters. We added table 1, which represent the DH parameters of Pepper required to use the Python Robotic Toolbox. Other calculation details in the Python script are also added in lines  257 to 260 and in table 2 (motor torque constants of Pepper). 

Comment 2:

How do you detect that the user is holding the robot's hand/arm? Do you assume they are always in contact? Would it be possible for the calculation to estimate force even when the user is not holding the robot, causing the robot to wrongly navigate to a certain direction?

Response: To answer this comment we added the next sentences in section 3.4 to our manuscript:

The Pepper robot is equipped with tactile sensors located on its palms and fingertips. These tactile allow the robot to sense when someone is holding its hand. Once the sensors detect the pressure, they send signals to the robot's internal system enabling it to interpret the touch and respond appropriately. In this work, the control algorithms are executed only when the robot's internal system detects that the Pepper’s hand is touched. Therefore, the estimation of the force is not executed when the user is not holding the robot. As a result, force estimation is not performed when the user is not holding the robot, which prevents the robot from unintentionally navigating in an unexpected direction.

Comment 3:

The authors claim that the work is not intended to people with motion disabilities nor is it intended to support mobility. Therefore, I may assume that the solution is destined to healthy users. Since there was no limitation on the age of the possible users, walking speed may vary from a slower pace to a faster one. Did you perform any analysis on the maximum speed of the robot so that it would be possible to perceive if the robot is "slowing down" the human walking pace? It seems that the proposed solution would be more adequate to users with higher age, as the walking pace diminishes as the person gets older.

Response: To answer this comment we added the next sentences in the conclusion section of  our manuscript:

The walking speed of humans can vary based on factors like age, fitness level, terrain, and purpose of movement. While the maximum velocity of the Pepper robot may be slower than a healthy human's typical walking speed, it's crucial to consider that the robot's primary objective is not to replicate human locomotion but to engage with humans in social and service contexts. When walking hand in hand with other humans, the average velocity tends to be slower, influenced by individual walking speeds, physical condition, and comfort level. This slower pace offers opportunities for social bonding, conversations, shared experiences, and a sense of togetherness. Individuals naturally adjust their pace to accommodate slower walkers, ensuring the group remains connected. In this context, future research directions could investigate potential discomfort or impatience, as well as the impact on social bonding, by considering different walking velocities and examining how the Pepper robot's slowdown in walking pace during hand-in-hand interactions affects human partners of various age groups. 

Comment 4:

Another important point that may be already included in the authors plans for future work (usability analysis) is the influence of Pepper's structure in the walking activity. Due to the format of its base, the user's feet/legs may touch the robot's base causing some accident. One possible solution is to increase the shoulder angle to make the robot a little bit more distant to the user. Do you believe is there any impact in the estimation of the forces if the user is holding the robot arm in a distance higher than the one showed on the demo video?

Response: To answer this comment we added the next sentences in the conclusion section:

Another important aspect to consider in future work is the influence of Pepper's structure on the walking activity. This includes not only the shoulder angle but also the size and height of individuals. Children, with their shorter height and smaller size, are more prone to potential collisions with the robot's base during walking. They may encounter challenges in maintaining a safe distance and may not fully comprehend the associated risks. Additionally, children tend to have less awareness of their surroundings and exhibit more unpredictable movements, further increasing the likelihood of accidental contact with the robot's base. To mitigate these collision risks, the utilization of Pepper's range sensors becomes crucial. By leveraging the range sensor data, the robot can effectively detect obstacles and potential collision hazards, including objects and body parts that may come into contact with its base. This enables the robot to adapt its walking trajectory, slow down, or stop altogether when necessary to avoid accidents.

Comment 5:

 More general comments and minor errors are listed as follows.

"an Huma-Robot" -> "a Human-Robot"

"use, the topic" -> "use the topic"

"Moreover, 7" -> "Moreover, Figure 7"

" figure 8 objects" -> " figure 8, objects"

Response:

We fixed these minor errors as suggested by the reviewer.

Reviewer 3 Report

General comment: the study is presented as a pilot study aimed at proposing an interesting system architecture that could allow the Pepper robot to walk close to the person, holding his or her hand, while also taking into account the surrounding environment. The research seems very promising, physical activity indeed being one of the key components that ensure the maintenance of a high quality of life, although with the obvious limitations of a pilot study.

The article seems to me to be well written, I just have a few minor points to make to make the article easier to read. In general, the major comments relate to the introduction, which I think should be stated more clearly and neatly, to allow for a smoother reading and make the purpose of the study more clear. Also, I would expand the experimental validation part, adding information. The numbers refer to the article line.

Introduction:

22: The authors could make the concept of "mental health" more explicit, explaining in what sense it is important in maintaining a high quality of life for people.

23: Reference is made to the use of robots in the field of mental health. The authors could expand this concept by adding some work describing such uses to make the concept, itself very interesting, more concrete.

26: Include a literature reference that supports this statement

33: The concept of "exercise" is introduced at this point, without then making a connection with the following sentences, where the concept of solitude is taken up instead. I would therefore first give an overview of the concept of "loneliness," and then move on to introduce the notion of "exercise."

44: It would be helpful to explain what these usability issues might be, so that the connection with the concept of social robots, the advantage of which is that they are more accessible, is smoother and more immediate. In this regard, the authors might find interesting insights in Bevilacqua, R. et al. (2015). Robot-Era Project: Preliminary Results on the System Usability. In: Marcus, A. (eds) Design, User Experience, and Usability: Interactive Experience Design. DUXU 2015. Lecture Notes in Computer Science(), vol 9188. Springer, Cham. https://doi.org/10.1007/978-3-319-20889-3_51, an interesting study of social robots with a focus on evaluating system usability.

49: The inclusion of the concept of "physical activity" here is not clear: this is introduced first [moderate exercise plays a significant role in maintaining a healthy lifestyle], then the concept of robotics and social robots is introduced, and then the idea of physical activity is taken up again. To make the reading smoother, please ask the authors to fix this part. It is clear what is meant, it could be better expounded.

50: Be careful in expressing a concept of this magnitude: walking is not always an accessible exercise for everyone, there are obviously physical limitations, and not always those who avoid walking do so out of boredom or anxiety. The authors are requested to rephrase and expand on this statement to avoid ignoring any physical issues and oversimplifying the activity of walking.

52: Authors are requested to bring back a reference that can support this statement.

Related work and contributions:

76: The authors could expand on the important concept of motivation. How might using the app motivate people to walk more? Are there, for example, any studies on this that can be cited to support this?

101: Reference is made to different contexts in which the Pepper robot is used. Just for completeness of exposition, it would be interesting to briefly describe these contexts

Experimental validation

254: One information I think is useful to know is the test participants: were participants from the general population recruited? If so, were specific criteria followed? The authors are requested to give more information in this regard.

255: It would be helpful to include a description of the test setting. For example, reference is made to a "public space," the authors could be more specific and give a brief description of the place where the test was conducted, so as to give the reader a better idea. I know that there is a video available, but the reader may not have a chance to view it in the immediate future.

255: Finally, it might be helpful to give an indication about the duration of the test and data collection.

Limitations

315: Just a thought: probably an additional limitation, which could then be a future development, is the type of soil on which Pepper was tested in this case. How might the results change based on soil variations? It might be interesting to include this type of reflection as a limitation.

Author Response

General comment: the study is presented as a pilot study aimed at proposing an interesting system architecture that could allow the Pepper robot to walk close to the person, holding his or her hand, while also taking into account the surrounding environment. The research seems very promising, physical activity indeed being one of the key components that ensure the maintenance of a high quality of life, although with the obvious limitations of a pilot study.

The article seems to me to be well written, I just have a few minor points to make to make the article easier to read. In general, the major comments relate to the introduction, which I think should be stated more clearly and neatly, to allow for a smoother reading and make the purpose of the study more clear. Also, I would expand the experimental validation part, adding information. The numbers refer to the article line.

Author’s response to the general comment: We sincerely thank the reviewer for providing valuable feedback on our manuscript.

Comment 1:

Introduction:

22: The authors could make the concept of "mental health" more explicit, explaining in what sense it is important in maintaining a high quality of life for people.

Response:  We aded the next sentence to anwer this comment:

Mental Health is a state of overall psychological well-being [12]. Mental health encompasses all aspects of human cognitive functioning, including thoughts, ideas, motivations, and directions that originate from the mind. These elements have a profound impact on how individuals communicate, behave, and function in both personal and social contexts. By prioritizing mental health, individuals can optimize their well-being and lead fulfilling lives [13]

[12]  Sharma, Urvashi, and Ravindra Kumar. "Positivity in mental and physical health." The International Journal of Indian Psychology, Volume 2, Issue 2, No. 2 (2015): 65.

[13] Nara, Kuldeep. "Study of mental health among sportspersons." International Journal of Physical Education, Sports and Health. 2017c 4.1 (2017): 34-37.

Comment 2:

23: Reference is made to the use of robots in the field of mental health. The authors could expand this concept by adding some work describing such uses to make the concept, itself very interesting, more concrete.

Response:  We added the next sentence to anwer this comment, which adds relevant references to related works centered in the improvement of mental helath using robotics systems:

Therefore, in recent years, the use of robots to address mental health issues has gained significant attention, particularly to provide solutions that help to mitigate the increase of mental  health issues, such as autism [14], dementia [15], and social anxiety [16]. Relevant and recent literature reviews in this area has been presented in [17 –19 ].

[14] Kim, Elizabeth S., et al. "Social robots as embedded reinforcers of social behavior in children with autism." Journal of autism and developmental disorders 43 (2013): 1038-1049

[15] Šabanović, Selma, et al. "PARO robot affects diverse interaction modalities in group sensory therapy for older adults with dementia." 2013 IEEE 13th international conference on rehabilitation robotics (ICORR). IEEE, 2013.

[16] Rasouli, Samira, et al. "Potential applications of social robots in robot-assisted interventions for social anxiety." International Journal of Social Robotics 14.5 (2022): 1-32.

[17] Riek, Laurel D. "Robotics technology in mental health care." Artificial intelligence in behavioral and mental health care. Academic Press, 2016. 185-203.

[18] Kabacińska, Katarzyna, Tony J. Prescott, and Julie M. Robillard. "Socially assistive robots as mental health interventions for children: a scoping review." International Journal of Social Robotics 13 (2021): 919-935.

[19] Scoglio, Arielle AJ, et al. "Use of social robots in mental health and well-being research: systematic review." Journal of medical Internet research 21.7 (2019): e13322.

Comment 3:

26: Include a literature reference that supports this statement

Response: We added the next references supporting the next statement: 

In this context, one of the primary drivers behind this trend is the growing sense of social isolation and loneliness that many individuals are experiencing [17,20,21].

[17] Riek, Laurel D. "Robotics technology in mental health care." Artificial intelligence in behavioral and mental health care. Academic Press, 2016. 185-203.

[20] Odekerken-Schröder, Gaby, et al. "Mitigating loneliness with companion robots in the COVID-19 pandemic and beyond: an integrative framework and research agenda." Journal of Service Management 31.6 (2020): 1149-1162.

[21] Gasteiger, Norina, et al. "Friends from the future: a scoping review of research into robots and computer agents to combat loneliness in older people." Clinical interventions in aging (2021): 941-971.

Comment 4:

33: The concept of "exercise" is introduced at this point, without then making a connection with the following sentences, where the concept of solitude is taken up instead. I would therefore first give an overview of the concept of "loneliness," and then move on to introduce the notion of "exercise."

Response: We configure the explanation according to the suggestion of the reviewer (first giving an overview about the negative issues due to loneliness and then introducing  physical activity or exercise as a possible solution to mitigate these negative issues): 

This loneliness can exacerbate self-destruction, especially by losing previously active roles and assuming more passive positions [23]. This can lead to a loss of motivation to accomplish activities of daily living (ADL), which can lead to unsafe circumstances and reduced quality of life [24]. However, loneliness is not limited to older adults. Younger generations have also felt the impact of social isolation and disrupted routines. This issue is boosted by the rise in single-person households in countries such as Korea and Japan [25]. As the loneliness pandemic continues to affect all-ages individuals worldwide, the search for technological solutions that can help alleviate its negative mental and physical effects will become increasingly relevant. One possible solution is the use of regular physical activity or exercise to mitigate negative these effects. In particular, moderate exercise plays a significant role in maintaining a healthy lifestyle by offering a range of physical and mental health benefits by releasing endorphins, elevating mood and improving mental well-being [26,27]. Therefore, the development of technological solutions helping to establish a routine and sense of purpose through exercise can potentially combat feelings of isolation, resulting in a safer and more fulfilling life.

Comment 5:

44: It would be helpful to explain what these usability issues might be, so that the connection with the concept of social robots, the advantage of which is that they are more accessible, is smoother and more immediate. In this regard, the authors might find interesting insights in Bevilacqua, R. et al. (2015). Robot-Era Project: Preliminary Results on the System Usability. In: Marcus, A. (eds) Design, User Experience, and Usability: Interactive Experience Design. DUXU 2015. Lecture Notes in Computer Science(), vol 9188. Springer, Cham. https://doi.org/10.1007/978-3-319-20889-3_51, an interesting study of social robots with a focus on evaluating system usability.

Response: We added the next statement that lists some of the factors or challenges related to adoption of the robotics technology as references where thees concepts are dezribed in depth, including the references suggested by the reviewer

These factors or challenges, such as acceptance, attitudes, user-friendliness, trust, utility, safety, anxiety, perceived intelligence, social presence, human-likeness, aesthetics, has been described and reviewed in [30-33].

[30] Dino, Michael Joseph S., et al. "Nursing and human-computer interaction in healthcare robots for older people: An integrative review." International Journal of Nursing Studies Advances (2022): 100072.

[31] Naneva, Stanislava, et al. "A systematic review of attitudes, anxiety, acceptance, and trust towards social robots." International Journal of Social Robotics 12.6 (2020): 1179-1201.

[32] Coronado, Enrique, et al. "Evaluating quality in human-robot interaction: A systematic search and classification of performance and human-centered factors, measures and metrics towards an industry 5.0." Journal of Manufacturing Systems 63 (2022): 392-410.

[33] Bevilacqua, Roberta, et al. "Robot-era project: Preliminary results on the system usability." Design, User Experience, and Usability: Interactive Experience Design: 4th International Conference, DUXU 2015, Held as Part of HCI International 2015, Los Angeles, CA, USA, August 2-7, 2015, Proceedings, Part III 4. Springer International Publishing, 2015.

Comment 6:

49: The inclusion of the concept of "physical activity" here is not clear: this is introduced first [moderate exercise plays a significant role in maintaining a healthy lifestyle], then the concept of robotics and social robots is introduced, and then the idea of physical activity is taken up again. To make the reading smoother, please ask the authors to fix this part. It is clear what is meant, it could be better expounded.

Response: We agree with the reviewer. The concept is redundantly introduced again in that paragraph. To resolve this issue, we will delete the sentences reiterating the concept of "physical activity". This allows us to directly focus on the main objective of the article in that paragraph, avoiding the redundancy present in our previous version.

Comment 7:

50: Be careful in expressing a concept of this magnitude: walking is not always an accessible exercise for everyone, there are obviously physical limitations, and not always those who avoid walking do so out of boredom or anxiety. The authors are requested to rephrase and expand on this statement to avoid ignoring any physical issues and oversimplifying the activity of walking.

52: Authors are requested to bring back a reference that can support this statement.

Response:  These two issues are connected to the previous comment that included the paragraph from line 49 to 61 in the previous version of our manuscript. These issues  has been addressed in the same manner. By adopting this approach, potential misunderstandings, highlighted by the reviewer, are effectively avoided, ensuring a clearer understanding for readers.

Comment 8:

Related work and contributions:

76: The authors could expand on the important concept of motivation. How might using the app motivate people to walk more? Are there, for example, any studies on this that can be cited to support this?

Response: We have included references to articles that share similar objectives, which support the claim and highlight the importance of robotics systems in encouraging health activities, such as exercise. For this, we have incorporated the following sentences. 

The persuasive and motivational capabilities of robots are particularly relevant in various cases, especially those involving people's health recovery and maintenance [48]. Numerous studies have demonstrated the effectiveness and practicality of robotics systems in motivating people during rehabilitation [49,50] as well as in promoting healthy activities like exercise [51,52] across different age groups. The application discussed in this work shares similar objectives by developing a system architecture that can potentially be used to promote a healthy lifestyle through walking. 

[48] Baroni, Ilaria, et al. "Designing motivational robot: how robots might motivate children to eat fruits and vegetables." The 23rd IEEE International Symposium on Robot and Human Interactive Communication. IEEE, 2014.

[49] Colombo, Roberto, et al. "Design strategies to improve patient motivation during robot-aided rehabilitation." Journal of neuroengineering and rehabilitation 4 (2007): 1-12.

[50] Schneider, Sebastian, and Franz Kummert. "Motivational effects of acknowledging feedback from a socially assistive robot." Social Robotics: 8th International Conference, ICSR 2016, Kansas City, MO, USA, November 1-3, 2016 Proceedings 8. Springer International Publishing, 2016.

[51] Fasola, Juan, and Maja J. Mataric. "Using socially assistive human–robot interaction to motivate physical exercise for older adults." Proceedings of the IEEE 100.8 (2012): 2512-2526.

Comment 9:

101: Reference is made to different contexts in which the Pepper robot is used. Just for completeness of exposition, it would be interesting to briefly describe these contexts

Response: We have incorporated the following sentences and references:

Pepper has been designed to interact with humans in various contexts, such children robot interaction [8], rehabilitation care [55] and public spaces [56], making it a suitable candidate for enabling humans to walk hand-in-hand through pHRI.

[8] Coronado, Enrique, Xela Indurkhya, and Gentiane Venture. "Robots meet children, development of semi-autonomous control systems for children-robot interaction in the wild." 2019 IEEE 4th international conference on advanced robotics and mechatronics (ICARM). IEEE, 2019.

[55] Sato, Miki, et al. "Rehabilitation care with Pepper humanoid robot: A qualitative case study of older patients with schizophrenia and/or dementia in Japan." Enfermeria clinica 30 (2020): 32-36.

[56] Niemelä, Marketta, Päivi Heikkilä, and Hanna Lammi. "A social service robot in a shopping mall: expectations of the management, retailers and consumers." Proceedings of the Companion of the 2017 ACM/IEEE International Conference on human-robot interaction. 2017.

Comment 10:

Experimental validation

254: One information I think is useful to know is the test participants: were participants from the general population recruited? If so, were specific criteria followed? The authors are requested to give more information in this regard.

Response: To answer to this comment we added the next sentences in the section 4.3:

Since the main objective of this study is to demonstrate the technological suitability of the system, no participants were recruited for experiments. The system was tested by the authors of this article.  As a result, this research does not require consent from any individuals, as no human subjects (aside from the authors of this article) were involved in the study. The involvement and recruitment of external participants will be considered for future work, which can potentially involve the analysis of usability and user experience factors.

Comment 11:

255: It would be helpful to include a description of the test setting. For example, reference is made to a "public space," the authors could be more specific and give a brief description of the place where the test was conducted, so as to give the reader a better idea. I know that there is a video available, but the reader may not have a chance to view it in the immediate future.

255: Finally, it might be helpful to give an indication about the duration of the test and data collection.

Response: To answer to these comments we added the next sentences in section 4:

This section presents the experimental results from an initial test of our proposed system architecture conducted in an open and public coffee space room adjacent to the GVlab of the Tokyo University of Agriculture and Technology. As depicted in Figure 6, this public space is primarily obstacle-free. It is important to note that in this particular application, the human assumes control of the robot, and therefore, no obstacle avoidance features were implemented. A video proving the technical suitability of our proposed system architecture in a 3 minutes experiment is available online …

Comment 13:

Limitations

315: Just a thought: probably an additional limitation, which could then be a future development, is the type of soil on which Pepper was tested in this case. How might the results change based on soil variations? It might be interesting to include this type of reflection as a limitation.

Response: To answer to these comments we added the next sentences in section 4.3 Limitations:

Furthermore, these studies can be expanded to encompass various types of soil surfaces on which Pepper walks or different interaction scenarios.

Reviewer 4 Report

The paper proposes a force control method for a companion robot with obstacle recognition capabilities to walk side-by-side person while holding hands; increasing companionship and promoting physical and mental health. While using force control to operate a social robot, like Pepper, might be innovative; this method is very common in robot manipulation and mobile robots. Therefore, it is difficult to assess the contribution of this work to the field of HRI or robotics in general. Likewise, the novelty of integrating obstacle recognition for robot behavior was not clear since there was no related work to justify this research.

The author could further explain the motion model of the robot using force control, for example, robot actions to avoid obstacles, the robot’s direction and speed with relation to the force, filtering techniques when sensing motor currents, etc.

Regarding object recognition, it was not clear how many objects were added to the training machine nor what was the purpose of selecting particular objects nor the usability of this feature for users. The study described the results of the force control and objective recognition for the Pepper robot without adequate metrics or comparable variables. For example, obstacle recognition figures showed results with low resolution and inadequate contrast. I'd recommend providing the success rate to identify objects at different distances in a table. Also, at the end of line 290, the authors mentioned that the reliability of each object was fixed with a different threshold; however, no results were provided.

Overall, the study did not provide enough results and many limitations as the authors mentioned in Section 4.3. which is common for research in the development phase. However, for these reasons, I’d suggest addressing their own limitations to show findings with robust force control and object recognition capabilities for a social robot.

Author Response

General comment:

The paper proposes a force control method for a companion robot with obstacle recognition capabilities to walk side-by-side person while holding hands; increasing companionship and promoting physical and mental health. While using force control to operate a social robot, like Pepper, might be innovative; this method is very common in robot manipulation and mobile robots. Therefore, it is difficult to assess the contribution of this work to the field of HRI or robotics in general. Likewise, the novelty of integrating obstacle recognition for robot behavior was not clear since there was no related work to justify this research.

Overall, the study did not provide enough results and many limitations as the authors mentioned in Section 4.3. which is common for research in the development phase. However, for these reasons, I’d suggest addressing their own limitations to show findings with robust force control and object recognition capabilities for a social robot.

Response: Thank you for your comments. We would like to emphasize that in this initial implementation, the object recognition feature is not utilized for obstacle avoidance purposes. Instead, it is solely employed to enhance communication and interaction with humans. Therefore, a in depth analysis of object recognition capabilities is out of the scope of this article.

Comment 1:

The author could further explain the motion model of the robot using force control, for example, robot actions to avoid obstacles, the robot’s direction and speed with relation to the force, filtering techniques when sensing motor currents, etc.

Response: In response to the reviewer's suggestion, we have incorporated the following additional details that were available and relevant to mention:

  • Overview of the control system: Figure 1 with its explanation from lines 184 to 191.
  • More details about how objects detected influence the robot behaviors from lines 355 to 365.
  • The direction of applied force on the Pepper’s left hand and robot coordinate system (Figure 4)

Comment 2:

Regarding object recognition, it was not clear how many objects were added to the training machine nor what was the purpose of selecting particular objects nor the usability of this feature for users. The study described the results of the force control and objective recognition for the Pepper robot without adequate metrics or comparable variables. For example, obstacle recognition figures showed results with low resolution and inadequate contrast. I'd recommend providing the success rate to identify objects at different distances in a table. Also, at the end of line 290, the authors mentioned that the reliability of each object was fixed with a different threshold; however, no results were provided.

Response: Thank you for your comments. As mentioned in our general response, conducting an in-depth analysis of object recognition capabilities is beyond the scope of this article. The novelty of this work does not involve the training and analysis of specific object recognition methods or algorithms. Instead, we propose a system architecture that offers a user-friendly interface, allowing the loading of any pre-trained YOLO-v4 model using OpenCV and TensorFlow. Table 3 (added in this new version of the manuscript) presents the list of objects available for recognition using the selected pre-trained model. The choice of this specific pre-trained model was based on its inclusion of common objects found in indoor scenarios. Regarding the selection of the threshold, it can be adjusted through the proposed interface. However, it's important to note that these values may vary depending on different scenarios, camera resolutions, and illumination conditions. Thus, we consider these threshold values to be context-dependent and not crucial for reproducibility purposes. These arguments are explained from lines 349 to 354 in the new version of the manuscript.

Round 2

Reviewer 4 Report

The authors addressed the comments. No further edits needed.